# Coupling Fairness and Pruning in a Single Run: a Bi-level Optimization Perspective

## Abstract

Deep neural networks have demonstrated remarkable performance in various tasks. With a growing need for sparse deep learning, model compression techniques, especially pruning, have gained significant attention. However, conventional pruning techniques can inadvertently exacerbate algorithmic bias, resulting in unequal predictions. To address this, we define a fair pruning task where a sparse model is derived subject to fairness requirements. In particular, we propose a framework to jointly optimize the pruning mask and weight update processes with fairness constraints. This framework is engineered to compress models that maintain performance while ensuring fairness in a single execution. To this end, we formulate the fair pruning problem as a novel constrained bi-level optimization task and derive efficient and effective solving strategies. We design experiments spanning various datasets and settings to validate our proposed method. Our empirical analysis contrasts our framework with several mainstream pruning strategies, emphasizing our method's superiority in maintaining model fairness, performance, and efficiency.

## 1 Introduction

The great success of Artificial Intelligence applications in diverse domains owes much of its achievement to the remarkable capabilities of Deep Neural Networks (DNNs) (He et al., 2016; Dong et al., 2015). To achieve state-of-the-art performance, DNNs often require an immense number of parameters to capture complex relationships within data, which is both computationally and storage intensive. In the pursuit of high performance and efficiency, various DNN model compression techniques have been developed (Han et al., 2015; He et al., 2017; Guo et al., 2016; Molchanov et al., 2016). Although these methods achieve high performance in efficient ways, researchers find that model compression techniques, like pruning, can introduce or exacerbate societal bias (Hooker et al., 2019; Liebenwein et al., 2021; Iofinova et al., 2023). It is crucial to address these concerns to ensure fairness and trust in deploying compressed models into real worlds.

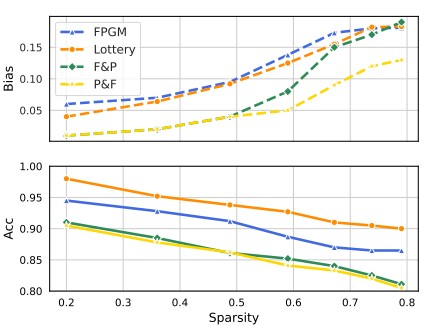

Figure 1: The accuracy and bias of different pruning methods and Prune & Fair patterns.

Addressing demographic disparity challenges in machine learning has received much attention in the recent literature on deep learning (Salvador et al., 2022; Wang et al., 2023; Zhu et al., 2023). The primary strategies include fairness-constrained optimization (Zafar et al., 2017) and adversarial machine learning (Madras et al., 2018). Notably, the majority of the current methodologies predominantly target dense deep neural networks that possess a vast parameter space. With a growing need for sparse deep learning, there is an emerging need to delve into the concept of fair pruning to harmonize the triad of performance, fairness, and efficiency. However, achieving this delicate balance in pruning is far from trivial. Network pruning is designed to identify an optimal mask for weights while ensuring high performance, whereas traditional fairness methods prioritize weight adjustments to reduce bias. This dilemma introduces challenges in simultaneously achieving fair-

ness and identifying suitable masks, given the intertwined relationship between masks, weights, and fairness constraints.

One intuitive approach to derive a fair and compressed model is by sequentially integrating fair learning and model pruning. Yet, this strategy presents several complications. Pruning a fair model prioritizes weights with the goal of maximizing accuracy, often overlooking fairness requirements. Furthermore, when one seeks to tackle this bias in the pruned model, there's a noticeable degradation in accuracy. In Fig. 1, we preliminary evaluate four methods at various sparsity levels: (1) the classic structured pruning method (FPGM) (He et al., 2019), (2) the classic unstructured pruning method (Lottery) (Frankle & Carbin, 2018), (3) building a fair model and subsequently pruning it (F&P), and (4) pruning a model and then fine-tuning it with fair constraints (P&F). The results show that pruning inevitably introduces bias while intuitive methods cannot address this challenge.

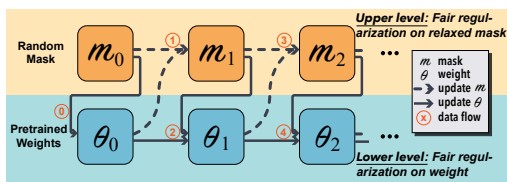

Figure 2: Sketch of the proposed framework.

To strike a harmonious balance among performance, efficiency, and fairness, a unified strategy is essential, considering the interactions among masks, weights, and constraints. In this work, We present a novel solution called **Bi**-level **F**air **P**runing (BiFP) based on bi-level optimization to ensure fairness in both the weights and the mask. Fig. 2 shows the overview of our proposed BiFP framework. We first initialize a mask $m_0$ and optimize the network weights with fairness constraints. Then, we fix the weights, and the relaxed masks are optimized under the consideration of fairness. By iteratively optimizing weights and masks under fairness constraints, our proposed approach effectively tackles the challenge and endeavors to preserve fairness throughout bi-level optimization processes. By **jointly optimizing the mask and weight** update processes while incorporating fairness constraints, we can strike a balance between preserving model performance and ensuring fairness in the pruning task. The proposed research represents a distinctive contribution towards achieving a harmonious balance between model efficiency and fairness. We summarize our contributions as follows.

- We comprehensively investigate the algorithmic bias issue in neural network pruning. We shed light on this unexplored task by comparing various pruning techniques and their potential to introduce bias.

- We introduce new fairness notions tailored specifically for model pruning tasks. These notions provide a more nuanced understanding of fairness in pruning.

- We pioneer a novel joint optimization framework that simultaneously promotes fairness in the mask and weight learning phases during model pruning. This unified approach ensures fairness and circumvents the need for multiple separate optimization runs.

- We perform extensive experiments that not only underline the effectiveness of our approach but also highlight its superiority over existing techniques in terms of fairness, performance, and efficiency. Through ablation studies, we explicitly demonstrate the indispensable role of both the mask constraint and weight constraint towards achieving fair pruning.

## 2 RELATED WORKS

### 2.1 FAIR CLASSIFICATION

Recently, algorithmic bias in machine learning has garnered significant attention from the research community, spawning a proliferation of methods and studies. Existing research tackles this challenge in two aspects: 1) defining and identifying bias in machine learning tasks, 2) developing bias elimination algorithms and solutions for conventional machine learning tasks.

To formally define fairness, various fairness notions have been proposed. The most well-recognized notion is *statistical parity*, which means the proportions of receiving favorable decisions for the protected and non-protected groups should be similar. The quantitative metrics derived from *statistical parity* include *risk difference*, *risk ratio*, *relative change*, and *odds ratio* (Zliobaite, 2017). Regarding bias elimination, existing methods are categorized into pre-processing, in-processing, and post-

processing. **Pre-processing** methods modify the training data to remove the potential prejudice and discrimination before model training. Common pre-processing methods include *Massaging* (Kamiran & Calders, 2009-02), *Reweighting* (Calders et al., 2009), and *Preferential Sampling*(Kamiran & Calders, 2012). The **in-processing** methods (Kamiran et al., 2010; Calders & Verwer, 2010; Zafar et al., 2017; Corbett-Davies et al., 2017) tweak the machine learning algorithms to ensure fair predictions, by adding fairness constraints or regularizers into the objective functions in machine learning tasks. The methods for **post-processing** (Kamiran et al., 2012; Hardt et al., 2016; Awasthi et al., 2020) correct the predictions produced by vanilla machine learning models.

## 2.2 NETWORK PRUNING

Network pruning is a technique aimed at reducing the size and computational complexity of deep learning networks. It involves selectively removing unnecessary weights or connections from a network while attempting to maintain its performance. Network pruning techniques can be broadly categorized into two main types. **Unstructured pruning** approaches remove certain weights or connections from the network without following any predetermined pattern. The early works (Han et al., 2015; Guo et al., 2016; Molchanov et al., 2016) remove weights from a neural network based on their magnitudes. The following works (Baykal et al., 2018; Lin et al., 2020; Yu et al., 2018; Molchanov et al., 2019) prune weights according to training data to approximate the influence of each parameter. Another group of researchers employs optimization methods to address the pruning problems (Ren et al., 2019; Zhang et al., 2018; 2022; Hoefler et al., 2021; Sehwag et al., 2020). **Structured pruning** involves pruning specific structures within the neural network, such as complete neurons, channels, or layers. Structured pruning frequently entails the removal of entire filters or feature maps from convolutional layers, resulting in a model architecture that is both structured and readily deployable. This pruning method has been explored extensively in numerous papers (Liu et al., 2019; Li et al., 2019; He et al., 2019; Dong et al., 2017; Ye et al., 2018; 2020). In filterwise pruning, filters are pruned by assigning an importance score based on weights (He et al., 2017; 2018) or are informed by data (Yu et al., 2018; Liebenwein et al., 2019).

## 2.3 FAIRNESS IN PRUNED NEURAL NETWORK

There has been increasing attention given to the possible emergence and exacerbation of biases in compressed sparse models, particularly as a result of model pruning processes. Paganini (2020) investigates the undesirable performance imbalances for a pruning process and provides a Pareto-based framework to insert fairness consideration into pruning processes. In the context of text datasets, Hansen & Søgaard (2021) provide an empirical analysis, scrutinizing the fairness of the lottery ticket extraction process. Tran et al. (2022) expands upon these insights by demonstrating how pruning might not only introduce biases but also amplify existing disparities in models. Exploring the domain of vision models, Iofinova et al. (2023) delved into the biases that might emerge in pruned models and proposed specific criteria to ascertain whether bias intensification is a probable outcome post-pruning. Lin et al. (2022) introduced a simple yet effective pruning methodology, termed Fairness-aware GRAdient Pruning mEthod (FairGRAPE), that minimizes the disproportionate impacts of pruning on different sub-groups, by selecting a subset of weights that maintain the fairness among multiple groups. Tang et al. (2023) offer a unique perspective to discern fair and accurate tickets right from randomly initialized weights. Despite the notable progress made in the realm of fair model compression, there is a lack of research that couples fair mask and fair weight learning processes in an efficient and simultaneous way.

## 3 PRELIMINARIES

In this section, we present foundational concepts to ensure a comprehensive understanding of the methodologies and analyses that follow. We begin by exploring the underlying principles of fair machine learning and then delve into the topic of deep neural network pruning.

### 3.1 FAIR CLASSIFICATION

Consider a dataset $\mathcal{D} = \left\{ (\mathbf{x}_i, \mathbf{y}_i, s_i) \right\}_{i=1}^{N}$, where $\mathbf{x}_i \in \mathcal{X}$ is an input feature, $\mathbf{y}_i \in \mathcal{Y}$ is a ground truth target and $s_i \in \mathcal{S}$ is a sensitive attribute, one can formulate a classification hypothesis space as

$f_\theta : \mathcal{X} \to \mathcal{Y}$ parameterized by $\theta$. The goal of a classification task is to find an optimal parameter $\theta^*$ such that: $\theta^* = \arg\min_\theta \mathbb{L}\left(f_\theta(\mathbf{x}), \mathbf{y}\right)$ where $\mathbb{L} = \frac{1}{N} \sum_{i=1}^{N} \ell_c(f_\theta(\mathbf{x}_i), \mathbf{y}_i)$ is the loss function and $\ell_c$ is a surrogate function, such as a hinge function or a logit function (Bartlett et al., 2006).

To take fairness into consideration, it is conventional to adopt fairness constraints into the classic classification task. Specially, we consider two demographic groups $\mathcal{D}^+ = \left\{(\mathbf{x}_i, \mathbf{y}_i, s_i) | s_i = s^+\right\}$ and $\mathcal{D}^- = \left\{(\mathbf{x}_i, \mathbf{y}_i, s_i) | s_i = s^-\right\}$ denoting the favorable and unfavorable groups, such as the male and female groups in the income prediction task. Given a performance metric $\mathbb{M}(f, \mathcal{D})$, i.e., accuracy, we say that classifier $f_\theta$ is fair if the difference between two groups is minor, i.e.,

$$\mathbb{F}(f_\theta, \mathcal{D}; \mathbb{M}) = |\mathbb{M}(f_\theta, \mathcal{D}^+) - \mathbb{M}(f_\theta, \mathcal{D}^-)| \leq \tau, \tag{1}$$

where $\tau$ is the user-defined threshold.

Note that the metric $\mathbb{M}$ is tailored according to the specific requirements, goals, or nuances of the application domain. As Iofinova et al. (2023), it is appropriate to measure the accuracy difference with respect to race and gender in the facial identification task since even a moderate difference in accuracy can lead to discrimination in real-world settings. Thus, this paper adopts the disparity of accuracy as the fairness metric. Thus, the metric $\mathbb{M}$ is defined as:

$$\mathbb{A}(f_\theta, \mathcal{D}) = \frac{\sum_{i=1}^{N} \mathbb{1}(\mathbf{y}_i = f_\theta(\mathbf{x}_i))}{|\mathcal{D}|}, \tag{2}$$

where $\mathbb{1}$ is an indicator function.

Then, we can formulate the fairness-aware classification problem as follows:

$$\theta^* = \arg\min_\theta \mathbb{L}(f_\theta(\mathbf{x}), \mathbf{y}) \quad s.t. \quad \mathbb{F}(f_\theta, \mathcal{D}; \mathbb{A}) \leq \tau \tag{3}$$

## 3.2 Neural Network Pruning

The primary motivation behind network pruning is that many neural network parameters contribute negligibly to the final prediction accuracy. These redundant or insignificant parameters cause over-parameterization, which raises the amount of computation needed during the inference and training phases. Network pruning seeks to mitigate this issue by identifying and eliminating such parameters, thereby achieving a more compact and optimized model.

Pruning techniques try to find a sparse structure (mask $m$) of a given model $f$ parameterized by $\theta$ and have no harm to its accuracy. Then we can formulate it as follows:

$$m = \arg\min_m \frac{1}{n} \sum_{i=1}^{n} \ell_c(f_{m \odot \theta}(\mathbf{x}_i), \mathbf{y}_i) \quad s.t. \quad ||m||_0 \leq k, \tag{4}$$

where $k$ is the desired remaining parameters. $m$ represents a binary mask designed to set specific parameters to a value of 0. $\|\cdot\|_0$ denotes the number of non-zero elements. The operator $\odot$ represents the element-wise product. Following the convention, we use sparsity to describe the ratio of zero parameters in a neural network, which can be computed by $1 - \frac{||m||_0}{||\theta||_0}$.

## 4 Fair Neural Network Pruning

Having established the foundational concepts of both fairness in machine learning and the intricacies of neural network pruning, we now converge on a critical intersection of these domains. The challenge lies in effectively integrating fairness considerations into the pruning process, ensuring that the efficiency gains of model compression do not inadvertently compromise the equitable performance of the model. In this section, we formulate the fair learning framework in the neural network pruning process and introduce a novel method that seamlessly marries these two domains, offering a pathway to achieve both compactness in neural networks and fairness in their predictions.

### 4.1 Fairness Notions for Compressed Models

Given any arbitrary compressed model, we aim to figure out whether this model produces fair and equalized results for demographic groups. One can measure the metric difference of the compressed

model on two demographic groups. If the compressed model has no difference regarding the selected metric, this compressed model is considered to achieve fairness in terms of the selected fairness notion. Formally, we use $f_c$ to represent the compressed model and formulate the fairness definitions as follows.

**Definition 1 (Performance Fairness)** *Given a compressed model $f_c$, we say the model is fair for performance if and only if its metric difference for two demographic groups is minor, i.e.,* $\mathbb{F}(f_c, \mathcal{D}; \mathbb{A}) = |\mathbb{A}(f_c, \mathcal{D}_{s+}) - \mathbb{A}(f_c, \mathcal{D}_{s-})| \leq \tau.$

In addition to the absolute measurement for the compressed model, we observe that the compressed model may amplify the bias compared to the original model $f$. The reason is that the model compression process introduces excess performance decrease for the unfavorable group. To capture the exacerbation bias induced by the model compression, we further introduce a metric for the accuracy shrinkage between the compressed model and the original model.

**Definition 2 (Compressed Model Performance Degradation)** *Given the original model $f$ and its compressed version $f_c$, the performance reduction is defined as:* $\mathbb{R}(f, f_c, \mathcal{D}) = \mathbb{A}(f, \mathcal{D}) - \mathbb{A}(f_c, \mathcal{D}).$

Further, we define a new fairness metric regarding the performance reduction of model compression.

**Definition 3 (Performance Degradation Fairness)** *Given the original model $f$ and its compressed version $f_c$, we say the models $f$ and $f_c$ are fair for performance degradation if and only if their metric decrease difference for two demographic groups is minor, i.e.,* $\mathbb{F}(f_c, \mathcal{D}; \mathbb{R}) = |\mathbb{R}(f, f_c, \mathcal{D}^+) - \mathbb{R}(f, f_c, \mathcal{D}^-)| \leq \tau.$

### 4.2 Fair and Efficient Learning via the Lens of Bi-level Optimization

Following our exploration of various fairness notions, we pivot toward the learning paradigms employed in the realm of fair and compressed neural networks. The intuitive integration of pruning, which focuses on applying masks to neural network weights, and fairness, which emphasizes adjustments to these weights, might seem trivial. In fact, the simplistic amalgamation might even undermine the objectives of both, as shown in Fig. 1, where both performance and fairness decrease. This drives us to reconsider the conventional methodology by designing a holistic solution. Our proposition is a simultaneous approach that enforces fairness in both masks and weights during the whole pruning phase. To this end, we cast the challenge as a bi-level optimization problem, and in the subsequent sections, we detail an efficient solution to this intricate puzzle.

Formally, neural network pruning aims to find a mask $m$ that determines the sparse pattern of the model, while fairness classification aims to acquire a fair parameter set $\theta$ that achieves fairness and maintains accuracy. To ensure the fairness requirement in both pruning masks and weights, we consider the model specified by an arbitrary mask $m$ and weights $\theta$ as $f_m : m \odot \theta; \mathcal{X} \rightarrow \mathcal{Y}$. Then, we formulate pruning and fairness classification as the following bi-level optimization problem:

$$\min \frac{1}{n} \sum_{i=1}^{n} \ell_c\big(f_{m \odot \theta^*(m)}(\mathbf{x}_i), \mathbf{y}_i\big), \tag{5}$$

$$s.t. \; \theta^*(m) = \arg\min_{\theta} \frac{1}{n} \sum_{i=1}^{n} \ell_c\big(f_{m \odot \theta}(\mathbf{x}_i), \mathbf{y}_i\big), \quad \mathbb{F}(f_{m \odot \theta}, \mathcal{D}; \mathbb{M}) \leq \tau$$

where $m$ and $\theta$ are the upper-level and lower-level optimization variables, respectively.

Bi-level optimization problems are sensitive with respect to the addition of constraints. Even adding a constraint that is not active at an optimal solution can drastically change the problem (Macal & Hurter, 1997). The constraint in Eq. 5 involving both the upper and the lower level variables can cause problems, i.e., the upper variable might not be able to accept certain optimal solutions of the lower level variable (Mersha & Dempe, 2006). In order to address this problem, we relax this constraint and discuss an efficient solution in the following subsection.

### 4.3 Convex Relaxation of Fairness Constraint

While we've posited a broad fairness constraint tailored for fair pruning, the generality of this constraint allows it to be effectively applied across a multitude of contexts. To shed light on its practical implications and to elucidate its inner workings, we delve into a specific example in this section. We consider the equalized accuracy constraint (Hardt et al., 2016) where the accuracy across different demographic groups is similar. i.e. $\mathbb{A}(f_{m \odot \theta}, \mathcal{D}^+) - \mathbb{A}(f_{m \odot \theta}, \mathcal{D}^-) \leq \tau$. To foster equalized accuracy constraint, we formulate the fairness constraint as follows:

$$\mathbb{F}_{\mathbb{A}} = \mathbb{E}_{X|S=1}[\mathbb{1}_{Y \cdot f_c(X) > 0}] - \mathbb{E}_{X|S=-1}[\mathbb{1}_{Y \cdot f_c(X) > 0}] \tag{6}$$

where $\mathbb{F}_{\mathbb{A}}$ is the abbreviation of $\mathbb{F}(f_{m \odot \theta}, \mathcal{D}; \mathbb{A})$, $\mathbb{1}(\cdot)$ is indicator function, and $f_c := f_{m \odot \theta}$. To ensure smooth and differentiable, we reformulate Eq. 6 as follows:

$$\mathbb{F}_{\mathbb{A}} = \mathbb{E}_{X|S=1}[\mathbb{1}_{Yf(X)>0}] - \mathbb{E}_{X|S=-1}[\mathbb{1}_{Yf(X)>0}] \tag{7}$$
$$= \mathbb{E}_X\left[\frac{P(S=1|X)}{P(S=1)}\mathbb{1}_{(Yf(X))>0}\right] + \mathbb{E}_X\left[\frac{P(S=-1|X)}{P(S=-1)}\mathbb{1}_{(Yf(X))<0}\right] - 1$$

The indicator function can be further replaced with the differentiable surrogate function $u(\cdot)$:

$$\mathbb{F}_{\mathbb{A}} = \frac{1}{N}\sum_{i=1}^{N}\frac{u\big(\mathbf{y}_i \cdot f_{m \odot \theta^*(m)}(\mathbf{x}_i)\big)}{P(S_i)} - 1 \tag{8}$$

Eq. 8 is convex and differentiable if the surrogate function $u$ is convex and differentiable at zero with $u'(0) > 0$ (Wu et al., 2019). In the general cases where $u$ is not convex, the differential nature of Eq. 8 ensures efficient optimization for gradient descent.

#### 4.3.1 Updating rule of the inner function.

After obtaining the differentiable variant for the fairness constraint, we refine the updating strategy for gradient descent of the inner function as follows:

$$\theta^t = \theta^{t-1} - \alpha \cdot \nabla_\theta\left(\frac{1}{n}\sum_{i=1}^{n}\ell_c + \mathbb{F}_{\mathbb{A}}\right)$$

Thus, the differentiable constraint can be seamlessly integrated with the inner optimization.

#### 4.3.2 Updating rule of the upper function.

For the upper optimization part, the masking variable is binary and discrete. We follow the conventional practice and relax the binary masking variables to continuous masking scores $m \in [0, 1]$ inspired by (Ramanujan et al., 2020). Then, we derive the gradient updating rule for the mask $m$ in the upper objective function:

$$grad(m) = \nabla_m \ell[\big(m \odot \theta^*(m)\big) + \mathbb{F}_{\mathbb{A}}] + \frac{d(\theta^*(m))}{dm}\nabla_\theta \ell[\big(m \odot \theta^*(m)\big) + \mathbb{F}_{\mathbb{A}}],$$

where $\nabla_m$ and $\nabla_\theta$ denote the partial derivatives of the bi-variate function $\ell(m \odot \theta)$. Then, we can update $m^t$ by following step:

$$m^t = m^{t-1} - \beta \cdot grad(m) \tag{9}$$

In this section, we define the fair learning task in neural network pruning and refine the bi-level optimization task to achieve fair and efficient pruning. Recognizing the challenges posed by the original formulation, we introduce a relaxation to accommodate convex and differentiable functions, greatly facilitating the optimization process. With these differentiable properties, we update the solving strategies intrinsic to bi-level optimization, ensuring their alignment with our tailored objectives.

## 5 Experiment

We evaluate our proposed method with several baselines in this section. The models are all implemented using PyTorch and evaluated with Intel(R) Core(TM) i9-10900X CPU and NVIDIA GeForce RTX 3070 GPU.

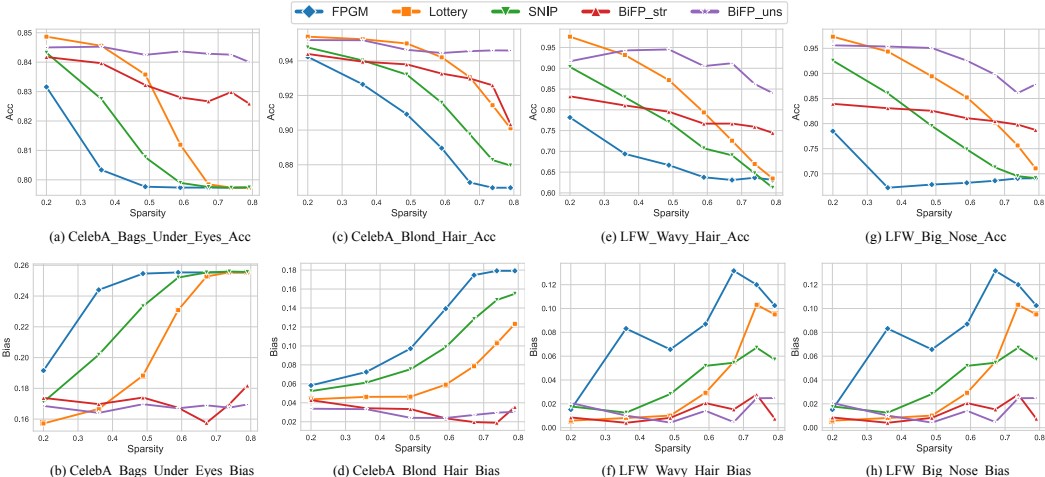

Figure 3: Accuracy and fairness evaluation of pruned **ResNet10** at varying sparsity levels. Subfig. (a, c, e, g) indicates the comparison of accuracy between the proposed method and the baseline methods. Subfig. (b, d, f, h) indicates the comparison of Compressed Model Fairness between the proposed method and the baseline methods. The proposed methods ensure fairness while having comparable even better accuracy.

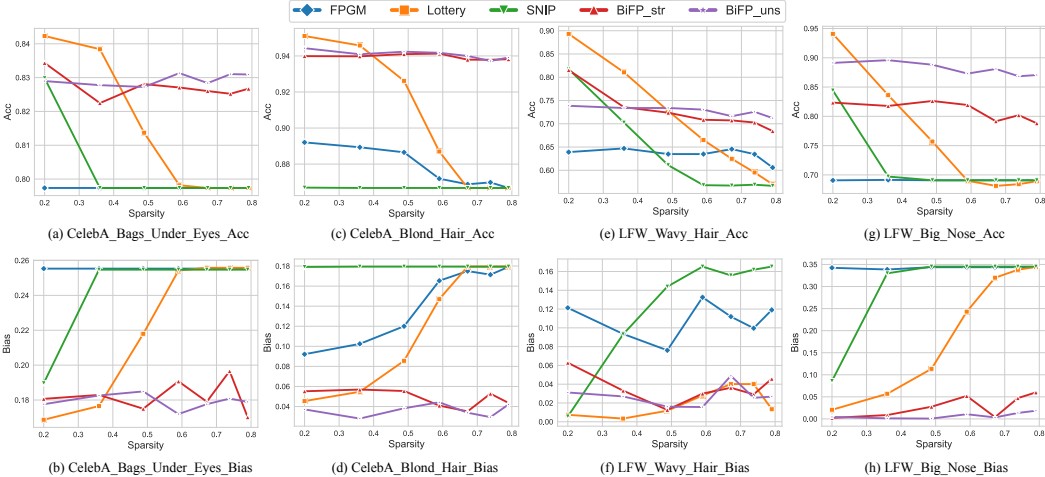

Figure 4: Accuracy and fairness of the pruned **Mobilenetv2** at varying sparsity level. Subfig. (a, c, e, g) indicates the comparison of accuracy between the proposed method and the baseline methods. Subfig. (b, d, f, h) indicates the comparison of Compressed Model Fairness between the proposed method and the baseline methods. The proposed methods ensure fairness while having comparable even better accuracy.

**Dataset.** We evaluate methods on two real-world image datasets, i.e., CelebA (Liu et al., 2015) and LFW (Huang et al., 2007). We adopt `Gender` as the sensitive attribute for both two datasets. For the CelebA dataset, we choose `Blond Hair` and `Bags-Under-Eyes` as our target attributes. For the LFW dataset, we choose `Wavy-Hair` and `Big-Nose` as our target attributes, respectively.

**Experiment Settings.** We pre-train a ResNet10 (He et al., 2016) network with the training set, prune and fine-tune with the validation set, and then report the accuracy and fairness violations at different sparsity in the test set. We strictly follow the sparsity ratio setting adopted by the Lottery Ticket Hypothesis (LTH) (Frankle & Carbin, 2018) to ensure a fair comparison.

**Methods.** We deploy the following baseline methods and separate them into two categories: unstructured pruning and structured pruning. Unstructured pruning methods contain the following methods: **Single-shot Network Pruning** (**SNIP**) calculates the connection sensitivity of edges and prunes those with low sensitivity. The structured pruning method contains **Filter Pruning via Geometric Median** (**FPGM**) pruning filters with the smallest geometric median. **The Lottery Ticket Hypothesis** (**Lottery**) states that there is a sparse subnetwork that can achieve similar test accuracy to the original dance network. It is worth pointing out that the proposed **Bi-level Fair Pruning** (**BiFP**) is capable of different pruning settings. We use **BiFP-str** and **BiFP-uns** to denote structured and unstructured variants, respectively.

### 5.1 PERFORMANCE AND FAIRNESS

We first perform experiments to show the capability of **BiFP** in mitigating bias during the model pruning. Fig. 3 and Fig. 4 compare the accuracy and bias evaluation of several methods on CelebA and LFW datasets. The first row of accuracy results shows that our methods (**BiFP-str** and **BiFP-uns**) are comparable to conventional pruning techniques with regard to model accuracy. The second row of results shows that conventional pruning techniques, including **FPGM**, **Lottery**, and **SNIP**, indeed introduce more bias with higher sparsity while the proposed methods, **BiFP-str** and **BiFP-uns**, consistently ensure fairness at any sparsity levels.

### 5.2 TRADE-OFF BETWEEN ACCURACY AND FAIRNESS

In the following experiments, we focus on the structured pruning variant of **BiFP** and drop the subscript for simplicity. We further investigate the trade-off between accuracy and fairness of the proposed method **BiFP** on CelebA with `Blond Hair` as our target attribute at the 70% sparsity level and report the results in Fig. 5. Through the trade-off curves, we conclude that the proposed method achieves a smaller bias compared with the baselines if we consider the same accuracy level (in a horizontal view of the plot). In another aspect, the proposed method achieves higher accuracy if we consider the same fairness level (in a vertical view of the plot). In a nutshell, the proposed method has a better trade-off for accuracy and fairness than the baselines.

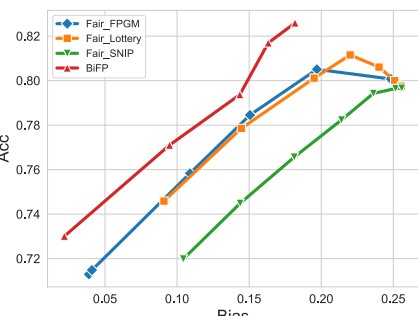

Figure 5: Trade-off between accuracy and fairness.

### 5.3 FAIRNESS AND TRAINING EFFICIENCY

To investigate the fairness capability and training efficiency of the proposed method and baselines, we amend the baselines with a two-stage fair training strategies. In the first stage, we prune a dense model using the conventional pruning techniques to get sparse models. In the second stage, we apply fairness constraint to retrain the parameters of sparse models to achieve the Performance Degradation Fairness. We record the number of training iterations for our proposed solutions and baseline methods and show them in Fig. 6. The red shade indicates the iterations used for **BiFP** to achieve the desired fairness and accuracy. In contrast, the grey shade demonstrates the iterations of the best baseline that has comparable performance to **BiFP**. The gap between the red shade and gray shade indicates that the proposed method is more efficient in achieving fairness. Our proposed method outperforms the best baseline methods while using notably less training cost, specifically savings 94.22% and 94.05% training iterations on the LFW and Celeba datasets.

### 5.4 ABLATION STUDIES

In the proposed method, we incorporated fairness constraints on the weights and masks of the network pruning simultaneously. To delve deeper into the implications of each component, we conducted an ablation study focusing on the fair weight and fair mask mechanisms separately. In particular, we remove the fair constraints for the mask in Eq. 5 and refer to this variant as **BiFP w/o m**. Similarly, we remove the fair constraints for the weights in Eq. 5 and refer to it as **BiFP w/o w**.

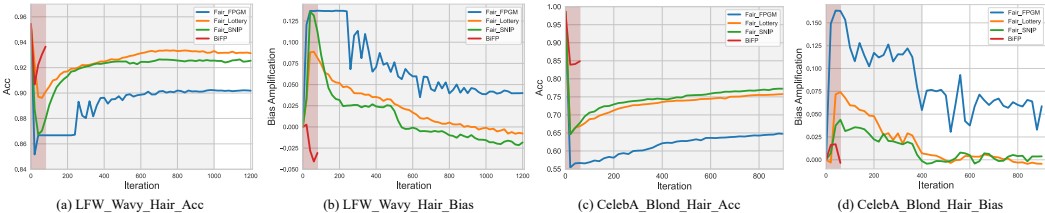

(a) LFW_Wavy_Hair_Acc  (b) LFW_Wavy_Hair_Bias  (c) CelebA_Blond_Hair_Acc  (d) CelebA_Blond_Hair_Bias

Figure 6: Training iterations used to obtain fair models using a two-stage pruning strategy. The red shade indicates the **BiFP** iterations while the grey shade indicates iterations of the best baseline.

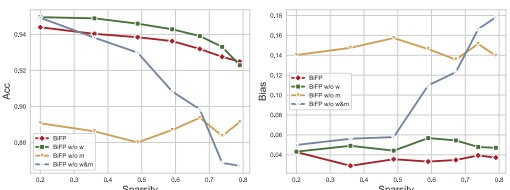
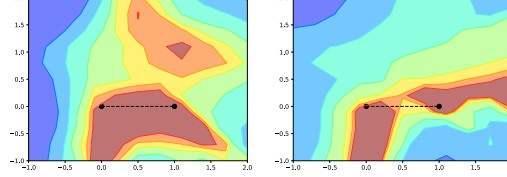

Figure 7: Ablation Studies on **BiFP**.

Figure 8: Interpolation of loss surfaces from **BiFP** to **BiFP w/o w** (left) & **BiFP w/o m** (right).

We further remove the fair constraints for both mask and weights, denoted by **BiFP w/o w&m**. The ablation study results are shown in Fig. 7. First, we observe that **BiFP w/o m** and **BiFP w/o w&m** have remarkably lower accuracy and higher bias than the original **BiFP**, indicating masks are necessary for both fairness and fairness. Although **BiFP w/o w** has better accuracy than the original **BiFP** at the low sparsity level, **BiFP** will outperform **BiFP w/o w** in terms of both accuracy and fairness at high sparsity level. The reason is **BiFP w/o w** is capable of preserving accuracy and maintaining fairness merely based on fair weights of the model. If the model is sparse, the weights cannot preserve accuracy and fairness anymore. Additionally, **BiFP w/o w** has relatively high accuracy but high bias as the weights are corrupted.

**Loss surface exploration.** Furthermore, we explore the implications of **BiFP** by exploring the loss surface of **BiFP** and its counterparts. Fig. 8 shows the loss surface lying between **BiFP** and **BiFP w/o w** or **BiFP w/o m**, respectively. The left plot indicates the loss surface from **BiFP BiFP w/o w** and the right plot indicates the loss surface from **BiFP** to **BiFP w/o m**. We use linear interpolation along the direction between **BiFP** and its counterpart. The right surface plot reveals a high-loss barrier (basin) between **BiFP** and **BiFP w/o m**, which demonstrates the fairness-constrained mask significantly ensures the performance in terms of both accuracy and fairness. It also suggests that only considering fairness-aware weights without fair masks, the optimization can get stuck at local minima that are isolated from better solutions. On the contrary, **BiFP** and **BiFP w/o w** (in the left plot) lie in the same basin, indicating a solution without fair weights is very close to the **BiFP** solution from the optimization perspective. The loss surfaces validate the ablation studies.

## 6  CONCLUSION

In this paper, we investigate the bias concerns in the neural network pruning task and mathematical formulate fairness notions for this task. Then, we introduce a novel constrained bi-level framework **BiFP** that seamlessly integrates fairness, accuracy, and model sparsity and enables the efficiency of pruning while maintaining both accuracy and fairness. We leverage the bi-level optimization to enforce fairness in both masks and weights simultaneously. In the extensive experiments, we show that the proposed solution is able to mitigate bias and ensure model performance with notably lower training costs.

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

# A APPENDIX

You may include other additional sections here.

