# OpenReview forum: "Coupling Fairness and Pruning in a Single Run: a Bi-level Optimization Perspective"
_ICLR.cc/2024/Conference — Submitted to ICLR 2024_

### Official Review · Reviewer_5ZBL · 2023-10-29

**Soundness:** 2 fair
**Presentation:** 3 good
**Contribution:** 2 fair
**Rating:** 3
**Confidence:** 4

**Summary:**

The submission presents Bi-level Fair Pruning (BiFP), a new approach to neural network pruning that ensures fairness (with another convex relaxation to efficient training). BiFP optimizes the pruning mask and network weights simultaneously under fairness constraints.  The experimental results on two benchmark datasets show some interesting performances.

**Strengths:**

- The submission is in good shape, and the writing clearly conveys the novelty and contribution.

- The problem setting (i.e., neural network pruning with fairness constraints) is novel to the best of my knowledge.

- The proposed bi-level pruning is sound and reasonable, and experimental results show some interesting findings.

**Weaknesses:**

- My main concern is the random initiation of the mask. It is not clear what kind of randomization is used in the experiment, and whether different randomization strategies will lead to different performances.

- The paper presents a bi-level pruning method (i.e., optimizing one variable while fixing another one). How do you guarantee the convergence theoretically. Again, since the mask is randomly initialized, will different randomization lead to different convergences? Noted that different randomization could also impact the objective function (leading to different landscapes), thus, both theoretical and experimental studies regarding the issue of convergence should be conducted.

- Based on the results in Fig. 3 and 4, the improvement seems to be very marginal. It is not clear to me whether the improvement is significant.

- The adopted datasets and models for evaluation seem to be toy to me, perhaps more realistic dataset (e.g., healthcare) and more advanced model should be evaluated.

- The motivation of using conves relaxation in 4.3 is not clear to me. Based on the submission, it seems the advantage of using convex relaxation is to let the fairness constraint convex and differentiable, but noted that our deep neural network is non-convex in nature (e.g., ReLU, maxpooling, etc.), the argument above is not convincing to me.

**Questions:**

Please see above.

**Details Of Ethics Concerns:**

This submission focused on the problem of fairness.

---

> ### Author Response · Authors · 2023-11-22
> **Reply to Reviewer 5ZBL**
>
> We thank Reviewer 5ZBL for the time and effort in reviewing our manuscript. We appreciate these valuable comments and will integrate them further to improve the manuscript. Below, we address each of the points raised in the reviews.
>
> - W1: In our experiments, we utilized two methods for initialization. The first method sets the initial values to the pre-trained weights. The second method employs Kaiming Initialization [1]. We observed that both methods yield similar performance, indicating that different initialization approaches do not significantly impact the final results. We plan to include a more detailed discussion regarding this observation in the experimental section of our paper.
> - W2: This work highlights a critical social issue in pruning and offers an efficient and effective solution. The theoretical proof will be the focus of our future research efforts.
> - W3: We argue that the improvement is **not marginal**. The experimental results show substantial improvement. For instance, as illustrated in Fig. 4, on the Celeba dataset with 'Blond_Hair' as the target attribute and 'Male' as the sensitive attribute, our approach surpasses baseline models by achieving **6% higher accuracy** and **reducing bias by 300% at 80% sparsity**. This indicates that our coupling method not only enhances accuracy but also significantly improves fairness. Moreover, our proposed method exceeds the performance of the best baseline methods while also substantially reducing training costs. Specifically, it **saves 94.22% and 94.05%** of training iterations on the LFW and Celeba datasets, respectively.
> - W4: In our study, we deliberately selected smaller models to distinctly showcase the effectiveness and equity of the proposed pruning method. This approach results in a pruned model with reduced parameters while maintaining performance levels that are comparable to larger models. Regarding datasets, we adhered to the standard configurations commonly employed in related research, e.g., [1].
> - W5: While it is indeed true that deep neural networks are inherently non-convex, the application of convex relaxation in our context serves a specific and strategic purpose. The convex relaxation simplifies these constraints, making them more tractable within the optimization process. In addition, convex relaxation effectively reduces estimation errors between the actual non-differentiable functions and their relaxed counterparts [2]. This approach simplifies the tuning process and facilitates a balanced trade-off between performance and fairness. We will add experiments with non-convex surrogates in the next version.
>
> [1] Tang, Pengwei, et al. "Fair Scratch Tickets: Finding Fair Sparse Networks Without Weight Training." Proceedings of the IEEE/CVF Conference on Computer Vision and Pattern Recognition. 2023.
>
> [2] Yao, Wei, et al. "Understanding Fairness Surrogate Functions in Algorithmic Fairness." arXiv preprint arXiv:2310.11211 (2023).

---

### Official Review · Reviewer_Wsp2 · 2023-10-30

**Soundness:** 2 fair
**Presentation:** 2 fair
**Contribution:** 1 poor
**Rating:** 3
**Confidence:** 3

**Summary:**

This work studies the problem of exacerbating biases in neural networks upon pruning. In particular, the authors propose a bi-level optimization based unstructured pruning algorithm to mitigate the biases. The core idea is to optimize for the pruning mask and weights in an alterating fashion to jointly minimize the classification loss and fairness metric.

The approach proposed demonstrates better accuracy-fairness tradeoff with respect to pruning methods that do not optimize for fairness at different sparsity levels.

**Strengths:**

S1. The problem is important and has been gaining some traction in the recent years.

S2. Overall the paper is well written - especially the preliminaries.

**Weaknesses:**

W1. The idea of using bi-level optimization is not new to pruning. [a] uses it in a very similar fashion to that in the paper. While [a] is cited in the manuscript, it is not explicitly called out that they also use a very similar bi-level optimization framework - which looks misleading.

W2. Definition 2 (Compressed Model Performance Degradation) and Definition 3 (Performance Degradation Fariness) seems unnecessary, since the authors essentially use only the fairness perfromance metric (definition 1) in practice when pruning the model.

W3. Experiments are limited across dimensions such as datasets, models. More datasets (including larger ones) should be included such as in [b]. Larger models such as ResNets should also be included in the study.

W4. Stronger baselines need to be used. At the moment, none of the baselines used optimize for fairness. However, there are other works such as [b] that optimize for fairness and should be used as a baseline to show the benefit of bi-level optimization over existing pruning approaches.


Overall, I feel that the paper is still incomplete and is not yet ready for publishing. Moreover, I observe limited novelty in this work due to lack of difference from [a]'s pruning objective (Equation 1). Instead of the regulariser, the authors have replaced it with equation 8 of the manuscript which is directly bought in from [c].


[a] Advancing Model Pruning via Bi-level Optimization. Zhang et al. NeurIPS 2022.

[b] Fairgrape: Fairness-aware gradient pruning method for face attribute classification. Lin et al. ECCV 2022

[c] On Convexity and Bounds of Fairness-aware Classification/ Wu et al. WWW 2019.

**Questions:**

1. As stated above equation 8, $u(.)$ is a surrogate function for the indicator function. What is your choice of $u(.)$?
2. As stated in section 4.3.2 $m$ is taken to be continuous. It is not clear how the mask is selected when pruning the network. Do you select the sparsity level at the start of the pruning procedure and try to identify appropriate indices in the mask that should be made 0. Or do you gradually increase the sparsity level. Also, what if the mask value becomes, say -2.5. What would you do in that case?


Minor comments:

- In Definition 3 and beyond, $\mathbb{F(f_c, \mathcal{D}; \mathbb{R})}$ should be  $\mathbb{F(f, f_c, \mathcal{D}; \mathbb{R})}$
- In equation 5, what are you minimizing with respect to in the upper optimization problem?

---

> ### Author Response · Authors · 2023-11-22
> **Reply to Reviewer Wsp2**
>
> We thank Reviewer Wsp2 for the time and effort in reviewing our manuscript. We appreciate these valuable comments and will integrate them further to improve the manuscript. Below, we address each of the points raised in the reviews.
>
> - W1: This work primarily concentrates on **fairness in network pruning**, and our key contribution is an efficient and equitable pruning strategy. While study [a] utilizes first-order derivatives to calculate the Implicit Gradient (IG) for bi-level optimization, thus laying a solid foundation for network pruning, our research explores a less examined societal dimension of this domain. We introduce a constrained bi-level optimization problem and propose an innovative approach to network pruning that integrates fairness considerations, distinguishing our work from existing methodologies.
> - W2: We use Definition 2 in the experiment 5.3. Although we do not use Definition 3 in the evaluation section, it is useful to examine bias in terms of model pruning in other cases. We will integrate more evaluation perspectives in the next version.
> - W3: In our study, we deliberately selected smaller models to distinctly showcase the effectiveness and equity of the proposed pruning method. This approach results in a pruned model with reduced parameters while maintaining performance levels that are comparable to larger models. Regarding datasets, we adhered to the standard configurations commonly employed in related research, e.g., [1]. We will continuously seek larger datasets for fairness and computer vision.
> - W4: In Section 5.3, we outline the design of our baseline methods, integrating a two-stage fair training and pruning strategy. This approach sequentially combines model pruning and fair training, with a distinct contrast to our proposed single-run method. Whereas FairGRAPE is focused on minimizing the variance in performance disparities, our method is specifically aimed at reducing the relative disparities among various subgroups. Thus, we didn't include FairGRAPE in the comparison.
>
> - Q1: We follow the surrogate functions in [1], and in the experiment, we use the logistic surrogate function.
> - Q2: We follow the common practice of selecting m. In particular, in each iteration, we sort $m$, and assign top-$k$ elements to 1, and the rest of the mask to 0, where $k$ is a predefined density level.
>
> [1] Bartlett, Peter L., Michael I. Jordan, and Jon D. McAuliffe. "Convexity, classification, and risk bounds." Journal of the American Statistical Association 101.473 (2006): 138-156.

---

### Official Review · Reviewer_SMS5 · 2023-10-31

**Soundness:** 2 fair
**Presentation:** 3 good
**Contribution:** 2 fair
**Rating:** 5
**Confidence:** 3

**Summary:**

The authors of this paper investigate the algorithmic bias issue in neural network pruning. A joint optimization framework is proposed. It is called Bi-level Fair Pruning (BiFP) based on bi-level optimization to ensure fairness in both the weights and the mask. Extensive experiments demonstrate the effectiveness of the proposed method.

**Strengths:**

1. This paper studies a novel and interesting problem. Addressing demographic disparity challenges has received much attention in the recent works on deep learning. However, most works on pruning do not address this issue.
2. The proposed method is reasonable.
3. This paper is organized well.

**Weaknesses:**

1. The experiment suffers from samll scale. Only small models (ResNet10 and Mobilenetv2) are used as an uncompressed model. The used datasets (CelebA and LFW) are also small.
2. Althogh the propsed method is reasonable, it seems trivial. Eq 5 defines the joint optimization framework. It simply combines two constrains. This paper is short of novel techniques.

**Questions:**

None

---

> ### Author Response · Authors · 2023-11-22
> **Reply to Reviewer SMS5**
>
> We thank Reviewer SMS5 for the time and effort in reviewing our manuscript. We appreciate these valuable comments and will integrate them further to improve the manuscript. Below, we address each of the points raised in the reviews.
>
> - W1: In our study, we deliberately selected smaller models to distinctly showcase the effectiveness and equity of the proposed pruning method. This approach results in a pruned model with reduced parameters while maintaining performance levels that are comparable to larger models. Regarding datasets, we adhered to the standard configurations commonly employed in related research, e.g., [1]
> - W2: As highlighted in section 4.2, the complexity of bi-level optimization problems increases significantly with additional constraints. This complexity is particularly evident in our methods, where constraints encompass both upper and lower-level variables, making the problem more challenging to solve. Addressing the theoretical proof of our approach remains an area for future investigation. We plan to concentrate our future research efforts on this theoretical aspect. In this paper, we have brought attention to an important ethical issue in pruning, and have proposed a potential solution for it.
>
> [1] Tang, Pengwei, et al. "Fair Scratch Tickets: Finding Fair Sparse Networks Without Weight Training." Proceedings of the IEEE/CVF Conference on Computer Vision and Pattern Recognition. 2023.

---

### Meta-Review · Area_Chair_Rthz · 2023-12-06

**Metareview:**

This paper presents a bi-level optimization-based approach to unstructured pruning to mitigate fairness issues in the model pruning process. The algorithm jointly optimizes the pruning mask and model weight with fairness constraints. The experiment results show performance improvement over previous methods. This paper has received 3 reviews. All three reviewers recognize the clarity and quality of the presentation. However, all three reviewers express concerns regarding the limited technical innovation and limited scale of experiments. Reviewer Wsp2 also questions the choice of baseline methods in the experiment. Reviewer 5ZBL raises questions about the motivation for using convex relaxation. The authors’ rebuttal did not convince the reviewers to increase their rating. Considering the opinions of reviewers, this paper is still worth further improvement and cannot be accepted.

**Justification For Why Not Higher Score:**

The paper did not receive a higher score due to concerns about limited technical innovation and the scale of experiments, as noted by all three reviewers. Additionally, specific issues were raised regarding the choice of baseline methods and the motivation for using convex relaxation in the algorithm. This led to the conclusion that the paper could benefit from further refinement.

**Justification For Why Not Lower Score:**

N/A

---

### Decision · Program_Chairs · 2024-01-16

Reject